

# A novel nomogram based on the patient's clinical data and CT signs to predict poor outcomes in AIS patients

Jingyao Yang[1], Fangfang Deng[2], Qian Zhang[2], Zhuyin Zhang[2],
Qinghua Luo[2] and Yeyu Xiao[2]

[1] Guangzhou University of Chinese Medicine, Guangzhou, China
[2] Department of Medical Imaging, Guangzhou Hospital of Integrated Traditional and Western Medicine, Guangzhou, China

Corresponding authors
Qinghua Luo,
luoqhzyy2023@163.com
Yeyu Xiao, xyyu73@163.com

## ABSTRACT

**Background:** The 2019 American Heart Association/American Stroke Association (AHA/ASA) guidelines strongly advise using non-contrast CT (NCCT) of the head as a mandatory test for all patients with suspected acute ischemic stroke (AIS) due to CT's advantages of affordability and speed of imaging. Therefore, our objective was to combine patient clinical data with head CT signs to create a nomogram to predict poor outcomes in AIS patients.

**Methods:** A retrospective analysis was conducted on 161 patients with acute ischemic stroke who underwent mechanical thrombectomy at the Guangzhou Hospital of Integrated Traditional and Western Medicine from January 2019 to June 2023. All patients were randomly assigned to either the training cohort ($n = 113$) or the validation cohort ($n = 48$) at a 7:3 ratio. According to the National Institute of Health Stroke Scale (NIHSS) score 7 days after mechanical thrombectomy, the patients were divided into the good outcome group (<15) and the poor outcome group ($\geq 15$). Predictive factors were selected through univariate analyses, LASSO regression analysis, and multivariate logistic regression analysis, followed by the construction of a nomogram predictive model. The receiver operating characteristic (ROC) curve was used to evaluate the predictive performance of the model, and bootstrapped ROC area under the curve (AUC) estimates were calculated to provide a more stable evaluation of the model's accuracy. The model's calibration performance was evaluated through the Hosmer-Lemeshow goodness-of-fit test and calibration plot, and the clinical effectiveness of the model was analyzed through decision curve analysis (DCA).

**Results:** Multivariate logistic regression analysis showed that hyperdense middle cerebral artery sign (HMCAS) (OR 9.113; 95% CI [1.945–42.708]; $P = 0.005$), the Alberta Stroke Program Early Computed Tomography Score (ASPECTS) > 6 (OR 7.707; 95% CI [2.201–26.991]; $P = 0.001$), NIHSS score (OR 1.085; 95% CI [1.009–1.166]; $P = 0.027$), age (OR 1.077; 95% CI [1.020–1.138]; $P = 0.008$) and white blood cell count (WBC) (OR 1.200; 95% CI [1.008–1.428]; $P = 0.040$) were independent risk factors for early poor outcomes after mechanical thrombectomy. The nomogram model was constructed based on the above factors. The training set achieved an AUC of 0.894, while the validation set had an AUC of 0.848. The bootstrapped ROC AUC estimates were 0.905 (95% CI [0.842–0.960]) for the training set and 0.848 (95% CI [0.689–0.972]) for the validation set. Results from the

Hosmer-Lemeshow goodness-of-fit test and calibration plot indicated consistent performance of the prediction model across both training and validation cohorts. Furthermore, the DCA curve demonstrated the model's favorable clinical practicality.

**Conclusion:** This study introduces a novel practical nomogram based on HMCAS, ASPECTS > 6, NIHSS score, age, and WBC that can well predict the probability of poor outcomes after MT in patients with AIS.

## INTRODUCTION

Acute ischemic stroke (AIS) is the abrupt occurrence of disturbed blood flow in the brain, which often results from the blockage of surrounding blood arteries by a blood clot. It is associated with high mortality and disability rates (*Ding et al., 2022*). Mechanical thrombectomy (MT), a common method of endovascular treatment, can remove emboli in a short time, restore the patency of blood vessels, and prevent the expansion of the ischemic area in a timely manner. In 2019, MT was recommended by the American Heart Association/American Stroke Association (AHA/ASA) guidelines as the standard treatment for patients with acute ischemic stroke (*Powers et al., 2019*). Although MT therapy is beneficial for patients with large vessel occlusion strokes, nearly half of these patients are unable to care for themselves 90 days after surgery, requiring assistance from others, and a greater 90-day mortality rate, according to published research (*Van Horn et al., 2021*; *Hassan et al., 2019*).

With the widespread use of mechanical thrombectomy, there are more and more studies exploring its prognostic factors. Predicting poor outcomes prior to thrombectomy is crucial. In addition to modifying treatment plans and enhancing acute stroke patients' prognosis, it can somewhat lessen the cost on society. MT is not the only way to treat people who have had an acute stroke. This surgery costs more than other ways to treat the condition. If poor outcomes for the patient can be predicted ahead of time, he can choose other treatments that may be more effective, and costs that are not necessary can be cut. Although more and more advanced imaging technologies (radiomics, multimodal MRI) are used to predict the prognosis of mechanical thrombectomy (*Xu et al., 2022*; *Zhou et al., 2020*), their long examination time and high cost severely limit the application of these checks. The use of non-contrast CT (NCCT) of the head as a mandatory assessment item for all patients with suspected AIS is strongly advised in the 2019 AHA/ASA guidelines (*Powers et al., 2019*). The benefits of NCCT to MRI include ease of use, affordability, and speedy imaging (*Lu et al., 2022*). NCCT remains the principal imaging modality for acute stroke evaluation in the majority of primary hospitals.

A nomogram is a prevalent statistical instrument in medicine, specifically for personalized risk prediction and prognosis evaluation. It signifies the magnitude of predictive factors *via* the length of line segments, so intuitively illustrating each factor's

contribution to the projected outcome. This depiction of feature weights enhances the interpretability of the prediction model, enabling users to thoroughly evaluate multiple elements in their decision-making process. In recent years, nomograms have been extensively utilized for prognostication in illnesses such as cancer, chronic renal disease, and pneumonia, facilitating clinical decision-making (*Tan et al., 2022*).

We only identified two existing nomogram models predicting poor outcomes after mechanical thrombectomy in AIS patients, created by *Zhang et al. (2022)* and *Li et al. (2021)*. Zhang et al. included the gender, collateral circulation, postoperative modified thrombolysis in cerebral infarction (mTICI), stroke-associated pneumonia, preoperative Na, and creatinine. *Li et al. (2021)* included age, the National Institute of Health Stroke Scale (NIHSS) score, and creatinine. The primary limitation of these two models is the inadequate gathering of predictive parameters prior to nomogram construction, as they excluded imaging data. Moreover, the model developed by Zhang et al. incorporated postoperative clinical information, hence hindering its capacity to assist physicians in preoperative decision-making.

Therefore, the purpose of this study was to combine clinical and preoperative NCCT data in order to develop and evaluate a nomogram prediction model for patients' early poor outcomes.

## MATERIALS AND METHODS

### Study design and data source

This retrospective cohort study was conducted following the guidelines of the Declaration of Helsinki and was approved by the Ethics Committee of Guangzhou Hospital of Integrated Traditional and Western Medicine (Ethical Application Ref: Guangzhou Hospital of Integrated Traditional and Western Medicine 20240325003). As this study was retrospective in nature, the requirement for written informed consent was waived by the review board.

This study recruited patients with acute ischemic stroke who underwent mechanical thrombectomy at Guangzhou Hospital of Integrated Traditional Chinese and Western Medicine between January 2019 and June 2023. The inclusion criteria were: (1) age $\geq$ 18 years old; (2) preoperative non-contrast computed tomography (NCCT) to exclude intracranial hemorrhage; (3) onset-to-treatment time < 24 h; (4) The modified Rankin Scale (mRS) score before the onset is $\leq$ 2 points; (5) The responsible vessel for cerebral infarction is the middle cerebral artery or internal carotid artery. The exclusion criteria were: (1) lack of complete clinical data. (2) hospital stay < 7 days.

### Clinical data collection

Demographics data (including age and gender), vascular risk factors (including hypertension, atrial fibrillation, diabetes, coronary disease and previous stroke), imaging findings on NCCT (including brain atrophy, basal ganglia calcification (BGC), leukoaraiosis, intracranial artery calcification (IAC), hyperdense middle cerebral artery sign (HMCAS), encephalomalacia, lacunar infarction, lenticular nucleus obscuration, insular ribbon sign and brain tissue swelling sign), baseline data (NIHSS score and the

Alberta Stroke Program Early Computed Tomography Score (ASPECTS), IV thrombolysis, location of lesion, admission systolic blood pressure (SBP) and diastolic blood pressure (DBP)) and laboratory data (including glucose level, urea, red blood cell (RBC) count, hemoglobin (HGB), white blood cell (WBC) count, platelets (PLT), sodium ion concentration, calcium ion concentration, potassium ion concentration, chloride ion concentration, blood creatinine (Crea), lactate dehydrogenase (LDH), lactate dehydrogenase isoenzymes (LDH1)). Patients were categorized into two groups based on their NIHSS score 7 days after MT, which included poor outcomes (NIHSS ≥ 15) and good outcomes groups (NIHSS < 15) (*Chalos et al., 2020*; *Kasner, 2006*). All the CT images were independently observed and recorded by a double-blind method by one attending physician (Jingyao Yang) and one deputy chief physician (Yeyu Xiao), each with more than 5 years and 10 years of experience in imaging diagnosis. When the evaluation results of the two doctors differed, the opinions were unified through consultation.

## Statistical analysis

Statistical analyses were conducted using SPSS (version 19; IBM Corp., Armonk, NY, USA) and the R Project for Statistical Computing (version 4.1.2). A *P*-value < 0.05 (two-sided) was considered statistically significant. Patients were randomly assigned to either the training cohort or the validation cohort at a 7:3 ratio. Categorical variables were presented as numbers (percentages), while continuous variables were reported as mean ± standard deviation (SD) or median (interquartile range (IQR)). Differences between the two groups were examined using the chisquare test for categorical variables and either the independent t-test or Mann-Whitney U test for continuous variables. The Shapiro-Wilk normality test was employed for continuous variables, and the independent sample t-test was utilized for measurement data that satisfied normal distribution assumptions.

Variables with *P* values < 0.05 were incorporated into collinearity analysis, and variance inflation factors (VIFs) and tolerances were calculated. Tolerance is a measure of the degree of collinearity between a particular independent variable and other independent variables, and it is the reciprocal of VIF. Tolerances > 0.1 or VIF < 5 denoted no discernible collinearity. The variables without significant collinearity were incorporated into least absolute shrinkage and selection operator (LASSO) regression. We selected significant predictors for inclusion in the multivariate logistic regression analysis and subsequent nomogram construction based on the findings of the LASSO regression and taking clinical relevance into consideration. Finally, the predictors were used to establish a nomogram prediction model. The area under the receiver operating characteristic (ROC) curve and the bootstrapped ROC area under the curve (AUC) are utilized to assess the discrimination of the prediction model. The model's calibration is assessed through the construction of a calibration curve and the use of the Hosmer-Lemeshow test. In the Hosmer-Lemeshow test, if *P* value > 0.05 indicates that the model exhibits a good fit. Meanwhile, decision curve

analysis (DCA) is applied to assess the model by computing the net benefit at each risk threshold and generating a DCA diagram.

## RESULTS

Figure 1 illustrates the flow chart that represents the process of selecting patients. From January 2019 to June 2023, a total of 219 patients diagnosed with ischemic stroke underwent MT at the Guangzhou Hospital of Integrated Traditional and Western Medicine. Through screening, a total of 161 patients were included in the study. Among them, there were 117 males and 44 females, aged 34 to 92 years old, with an average age of $(63.12 \pm 12.08)$ years. A total of 102 people had a good outcome, and 59 people had a poor outcome (Fig. 2). All of 161 included patients were randomly divided into a training set and a validation set in a ratio of 7:3. Statistical analysis revealed that there was no statistically significant disparity in variables between the two groups ($P < 0.05$) (Table S1). The data have been thoroughly randomized.

As shown in Table 1, in the univariate logistic analysis, the NIHSS score ($P < 0.001$), age ($P < 0.001$), ASPECTS ($P < 0.001$), ASPECTS > 6 ($P < 0.001$), leukoaraiosis ($P = 0.040$), brain atrophy ($P = 0.026$), BGC ($P = 0.025$), HMCAS ($P < 0.001$), WBC ($P = 0.011$), Crea ($P = 0.023$) were found to be significantly associated with poor outcome. A multicollinearity diagnosis was performed on the above 10 variables, and there was no significant collinearity among the variables (Table 2). The ideal value of $\lambda$ (lambda. min = 0.025) was determined using LASSO regression and tenfold cross-validation (Figs. 3A, 3B). The candidate characteristics were reduced to the following seven features with nonzero coefficients: the NIHSS score, age, ASPECTS > 6, BGC, HMCAS, WBC, Crea. The above seven variables were included in multi-factor logistic regression, and it was found that age (OR, 1.077; 95% CI [1.020–1.138]; $P = 0.008$), ASPECTS > 6 (OR, 7.707; 95% CI [2.201–26.991]; $P = 0.001$), NIHSS score (OR, 1.085; 95% CI [1.009–1.166]; $P = 0.027$), HMCAS (OR, 9.113; 95% CI [1.945–42.708]; $P = 0.005$), and WBC (OR, 1.200; 95% CI [1.008–1.428]; $P = 0.040$) are independent predictors of early poor outcomes after mechanical thrombectomy (Table 3).

A nomogram prediction model was established using the R language based on the five variables derived from multi-factor logistic regression (Fig. 4). Each variable corresponds to a score (0–100), and the total score is the sum of the scores of the five variables. The estimated probability of a poor outcome was obtained from the nomogram according to the total score. The nomogram demonstrated good discriminatory power, with an AUC of 0.894 (Fig. 5A) in the training cohort and an AUC of 0.848 (Fig. 5B) in the validation cohort. The Bootstrapped ROC AUC estimates were 0.905 (95% CI [0.842–0.960]) for the training set and 0.848 (95% CI [0.689–0.972]) for the validation set. The Hosmer-Leme show test showed good concordance between predicted and observed probability for the training cohort ($X^2 = 5.010$, df = 8, $P = 0.756$) and the validation cohort ($X^2 = 5.799$, df = 7, $P = 0.563$). The calibration plot also revealed significant predictive accuracy of the nomogram to predict poor outcomes after mechanical thrombectomy in the training

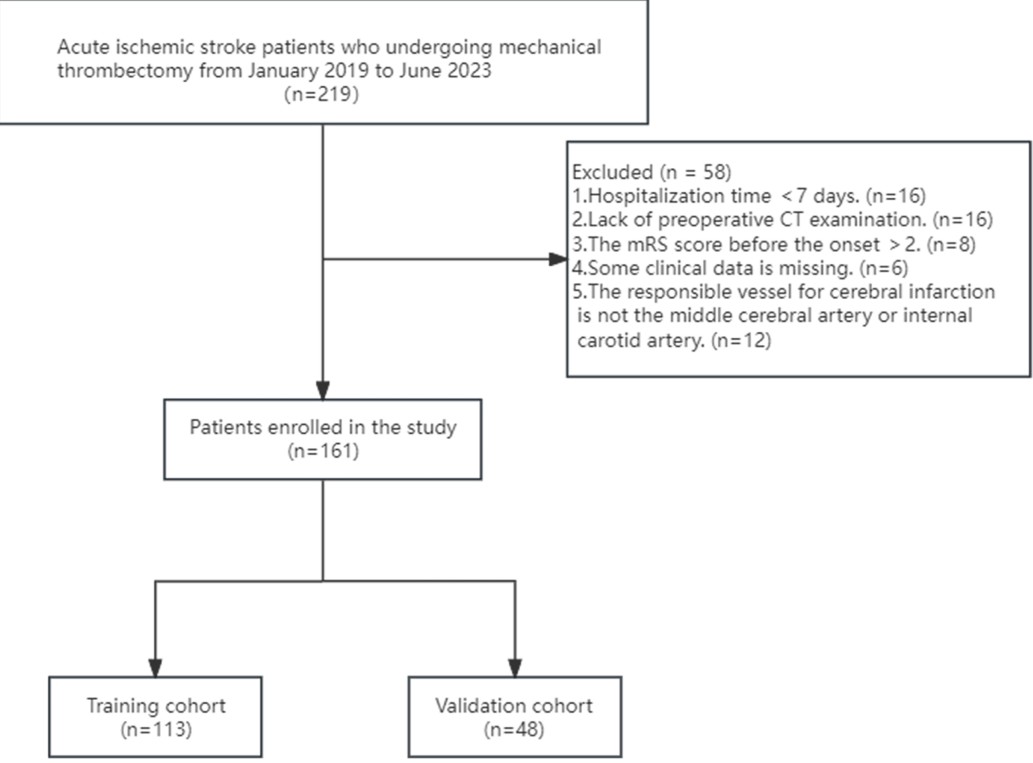

**Figure 1 Flowchart of patient inclusion and exclusion criteria.**

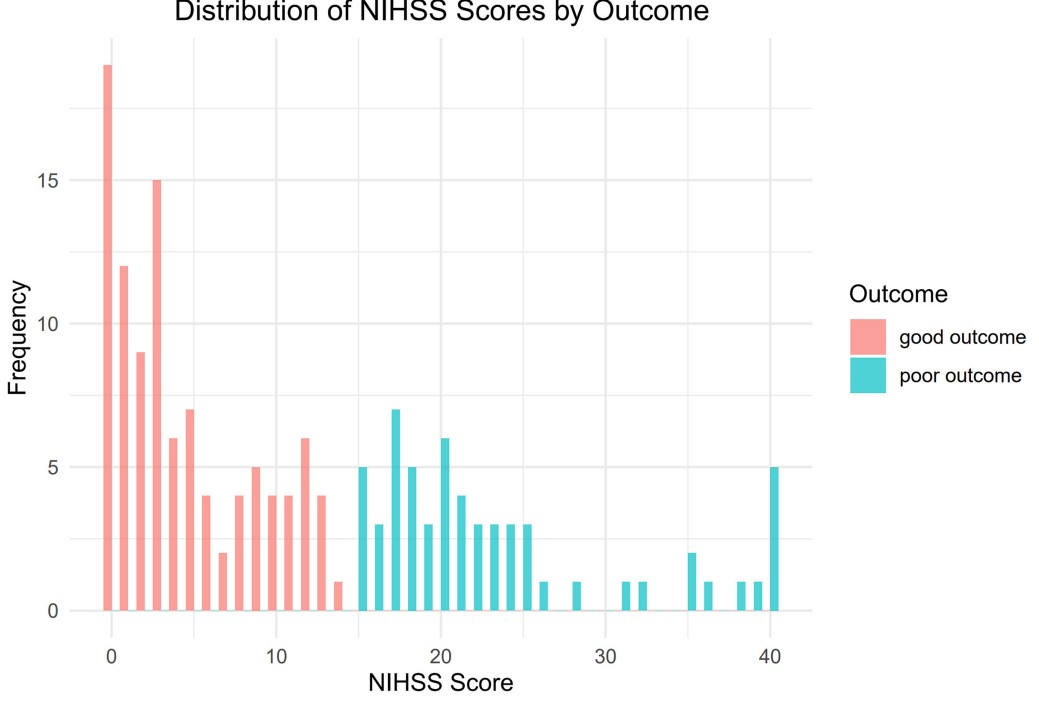

**Figure 2 Distribution of NIHSS scores by outcome.**

**Table 1 Comparison of demographics, imaging and clinical characteristics between patients with good and poor outcomes.**

| | Overall<br>N = 113 | Good outcomes<br>N = 71 | Poor outcomes<br>N = 42 | P-value |
|---|---|---|---|---|
| **Demographic data, n (%)** | | | | |
| Gender, male (%) | 81 (71.7) | 52 (73.2) | 29 (69.0) | 0.793 |
| Age, years (mean (SD)) | 63.33 (12.25) | 60.06 (11.71) | 68.86 (11.22) | <0.001 |
| **Vascular risk factors, n (%)** | | | | |
| Hypertension, yes (%) | 55 (48.7) | 31 (43.7) | 24 (57.1) | 0.234 |
| Diabetes, yes (%) | 24 (21.2) | 12 (16.9) | 12 (28.6) | 0.220 |
| Atrial fibrillation, yes (%) | 15 (13.3) | 9 (12.7) | 6 (14.3) | >0.999 |
| Previous stroke, yes (%) | 19 (16.8) | 12 (16.9) | 7 (16.7) | >0.999 |
| Coronary disease, yes (%) | 10 (8.8) | 5 (7.0) | 5 (11.9) | 0.591 |
| **Imaging findings, n (%)** | | | | |
| HMCAS, yes (%) | 19 (16.8) | 4 (5.6) | 15 (35.7) | <0.001 |
| BGC, yes (%) | 26 (23.0) | 11 (15.5) | 15 (35.7) | 0.025 |
| Leukoaraiosis, yes (%) | 39 (34.5) | 19 (26.8) | 20 (47.6) | 0.040 |
| Brain atrophy, yes (%) | 48 (42.5) | 24 (33.8) | 24 (57.1) | 0.026 |
| IAC, yes (%) | 22 (19.5) | 13 (18.3) | 9 (21.4) | 0.874 |
| Encephalomalacia, yes (%) | 20 (17.7) | 12 (16.9) | 8 (19.0) | 0.973 |
| Lacunar infarction, yes (%) | 63 (55.8) | 36 (50.7) | 27 (64.3) | 0.227 |
| Lenticular nucleus obscuration, yes (%) | 6 (5.3) | 4 (5.6) | 2 (4.8) | >0.999 |
| Insular ribbon sign, yes (%) | 8 (7.1) | 4 (5.6) | 4 (9.5) | 0.689 |
| Brain tissue swelling sign, yes (%) | 9 (8.0) | 4 (5.6) | 5 (11.9) | 0.406 |
| **Baseline data** | | | | |
| Location of lesion, right (%) | 58 (51.3) | 36 (50.7) | 22 (52.4) | >0.999 |
| ASPECT > 6, yes (%) | 85 (75.2) | 65 (91.5) | 20 (47.6) | <0.001 |
| ASPECT (median [IQR]) | 9.00 [8.00–10.00] | 10.00 [9.00–10.00] | 6.00 [6.00–9.75] | <0.001 |
| NIHSS score (median [IQR]) | 14.00 [9.00–21.00] | 11.00 [6.50–16.50] | 20.50 [15.00–24.00] | <0.001 |
| SBP, mmHg (mean (SD)) | 141.59 (23.69) | 139.48 (20.85) | 145.17 (27.75) | 0.219 |
| DBP, mmHg (mean (SD)) | 84.16 (12.82) | 84.10 (11.35) | 84.26 (15.14) | 0.948 |
| IV thrombolysis, yes (%) | 60 (53.1) | 38 (53.5) | 22 (52.4) | >0.999 |
| **Laboratory data** | | | | |
| NA, mmol/L (median [IQR]) | 140.00 [138.00–142.90] | 140.60 [138.30–142.85] | 140.00 [138.00–142.78] | 0.609 |
| K, mmol/L (median [IQR]) | 3.90 [3.60–4.14] | 3.89 [3.63–4.11] | 3.92 [3.58–4.16] | 0.753 |
| Ca, mmol/L (median [IQR]) | 2.24 [2.15–2.38] | 2.24 [2.15–2.38] | 2.26 [2.13–2.38] | 0.728 |
| Cl, mmol/L (median [IQR]) | 103.00 [101.00–105.00] | 103.50 [101.00–105.05] | 102.90 [100.93–104.97] | 0.915 |
| Crea, umol/L (median [IQR]) | 69.71 [61.20–93.08] | 68.31 [58.36–86.56] | 81.41 [67.01–106.17] | 0.023 |
| LDH, U/L (median [IQR]) | 222.00 [189.00–264.00] | 218.00 [189.50–256.50] | 237.00 [190.50–271.75] | 0.256 |
| LDH1, U/L (median [IQR]) | 30.00 [24.00–38.00] | 29.00 [23.50–35.50] | 32.00 [25.00–39.00] | 0.184 |
| HGB, g/L (median [IQR]) | 140.00 [126.00–151.00] | 141.00 [129.50–151.50] | 137.00 [118.25–149.50] | 0.124 |
| WBC, $\times 10^9$/L (median [IQR]) | 9.51 [7.76–11.38] | 8.60 [7.27–10.73] | 10.55 [8.49–12.49] | 0.011 |
| RBC, $\times 10^{12}$/L (median [IQR]) | 4.68 [4.32–5.11] | 4.69 [4.39–5.30] | 4.58 [4.20–4.93] | 0.069 |
| Urea, mmol/L (median [IQR]) | 5.62 [4.74–6.84] | 5.57 [4.59–6.71] | 6.01 [5.00–6.95] | 0.151 |
| Glu, mmol/L (median [IQR]) | 6.77 [5.98–8.83] | 6.58 [5.72–8.24] | 7.14 [6.36–9.19] | 0.225 |
| Plt, $\times 10^9$/L (median [IQR]) | 226.00 [181.00–270.00] | 226.00 [187.00–280.50] | 225.00 [175.00–265.50] | 0.607 |

**Table 2 Collinearity analysis of predictive factors.**

| Model | Unstandardized coefficients | | t | Sig. | Collinearity statistics | |
|---|---|---|---|---|---|---|
| | B | Std. error | | | Tolerance | VIF |
| (Constant) | −1.275 | 0.415 | −3.073 | 0.003 | | |
| Crea | 0.001 | 0.001 | 1.304 | 0.195 | 0.840 | 1.190 |
| HMCAS | 0.306 | 0.097 | 3.172 | 0.002 | 0.900 | 1.111 |
| Age | 0.009 | 0.004 | 2.588 | 0.011 | 0.609 | 1.643 |
| Wbc | 0.023 | 0.010 | 2.333 | 0.022 | 0.778 | 1.285 |
| ASPECTS > 6 | 0.556 | 0.171 | 3.250 | 0.002 | 0.215 | 4.652 |
| ASPECTS | 0.041 | 0.035 | 1.182 | 0.240 | 0.229 | 4.373 |
| NIHSS score | 0.012 | 0.005 | 2.491 | 0.014 | 0.791 | 1.265 |
| BGC | 0.064 | 0.086 | 0.743 | 0.459 | 0.898 | 1.114 |
| Leukoaraiosi | −0.143 | 0.113 | −1.260 | 0.210 | 0.405 | 2.469 |
| Brain atrophy | 0.055 | 0.108 | 0.505 | 0.614 | 0.412 | 2.425 |
| Dependent variable: outcome | | | | | | |

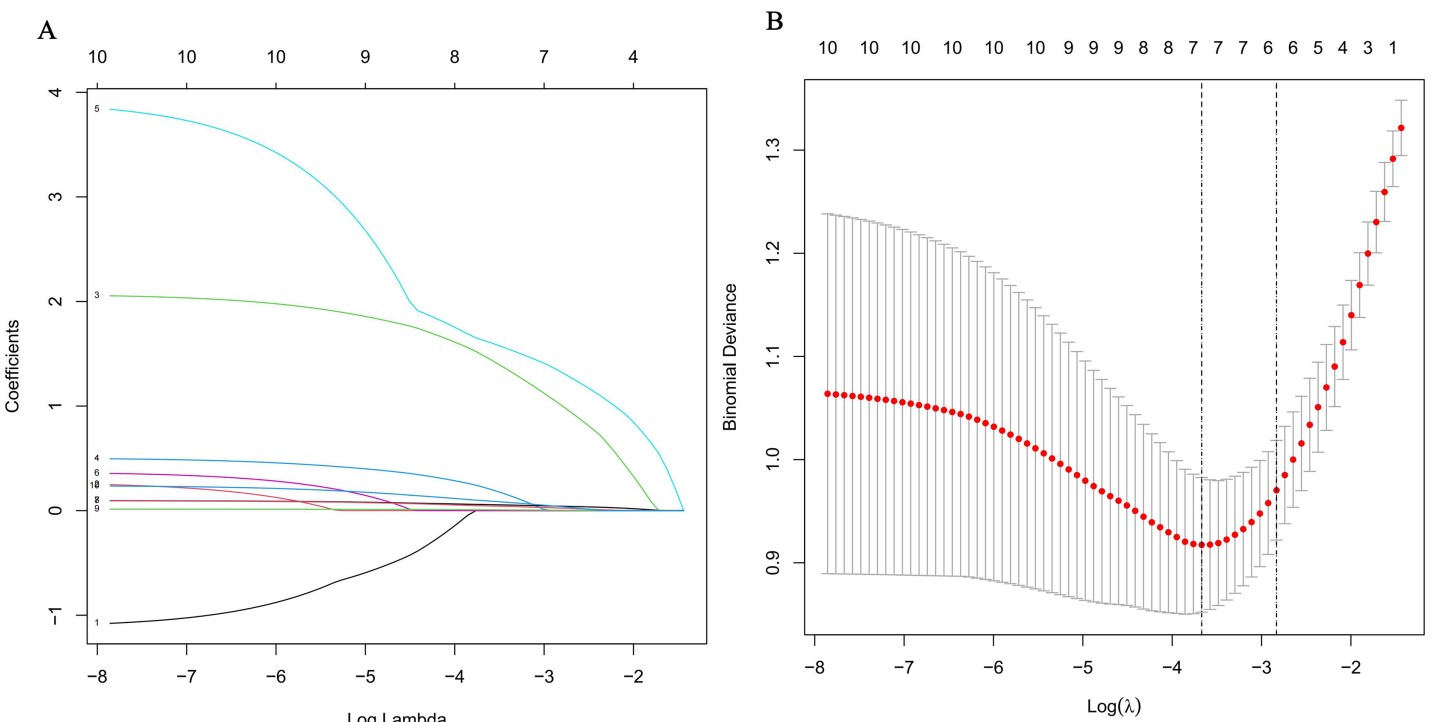

**Figure 3 Selection of predictors using the least absolute shrinkage and selection operator (LASSO) binary logistic regression model.** (A) Each curve in the figure illustrates the variation of each variable in relation to the coefficient. The ordinate represents the coefficient value, the lower abscissa denotes $\log(\lambda)$, and the higher abscissa indicates the count of non-zero coefficients in the model at this moment. (B) 10-fold cross-cross validation fitting and then selecting the model.

**Table 3 Multivariable logistic regression of possible predictors of poor outcomes in the training cohort.**

| Variable | β | SE | Wald χ² | P-value | OR | 95% CI |
|---|---|---|---|---|---|---|
| HMCAS | 2.210 | 0.788 | 7.862 | 0.005 | 9.113 | [1.945–42.708] |
| BGC | 0.518 | 0.643 | 0.649 | 0.421 | 1.678 | [0.476–5.918] |
| ASPECTS > 6 | 2.042 | 0.639 | 10.198 | 0.001 | 7.707 | [2.201–26.991] |
| NIHSS score | 0.081 | 0.370 | 4.888 | 0.027 | 1.085 | [1.009–1.166] |
| Age | 0.074 | 0.028 | 7.081 | 0.008 | 1.077 | [1.02–1.138] |
| Crea | 0.017 | 0.011 | 2.545 | 0.111 | 1.017 | [0.996–1.039] |
| WBC | 0.182 | 0.089 | 4.222 | 0.040 | 1.200 | [1.008–1.428] |

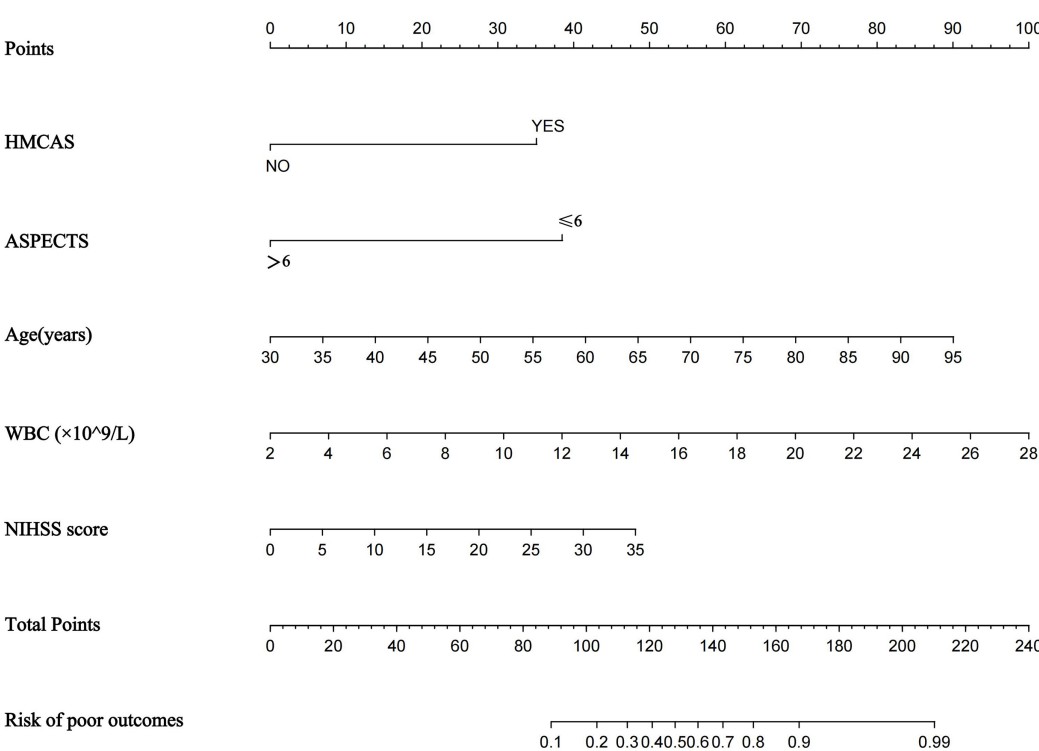

**Figure 4 Nomogram for predicting poor outcomes after mechanical thrombectomy.** The nomogram included age, ASPECTS > 6, WBC, HMCAS, and NIHSS score. The nomogram and associated algorithm are employed to forecast the risk of unfavorable outcomes in acute ischemic stroke with mechanical thrombectomy. Initially, identify the relevant score on the points line at the top of each variable for patients with AIS; then, aggregate all the scores and determine the equivalent point on the total points scale. Ultimately, ascertain the anticipated likelihood associated with the patient on the predicted value line. For example, a patient with the HMCAS, ASPECT > 6, 45 years old, WBC $10 \times 10^9$/L, baseline NIHSS scores of five would have a total of 113 scores. The probability of poor outcomes after mechanical thrombectomy was approximately 27% for the patient.

(Fig. 6A) and validation cohorts (Fig. 6B). The DCA illustrated that the newly developed nomogram showed a higher overall benefit in predicting poor outcomes compared to the 'all' or 'none' approaches. This was observed when the threshold probabilities fell within the range of 2.5% to 100.0% in the training group (Fig. 7A) and 13.5% to 100.0% in the

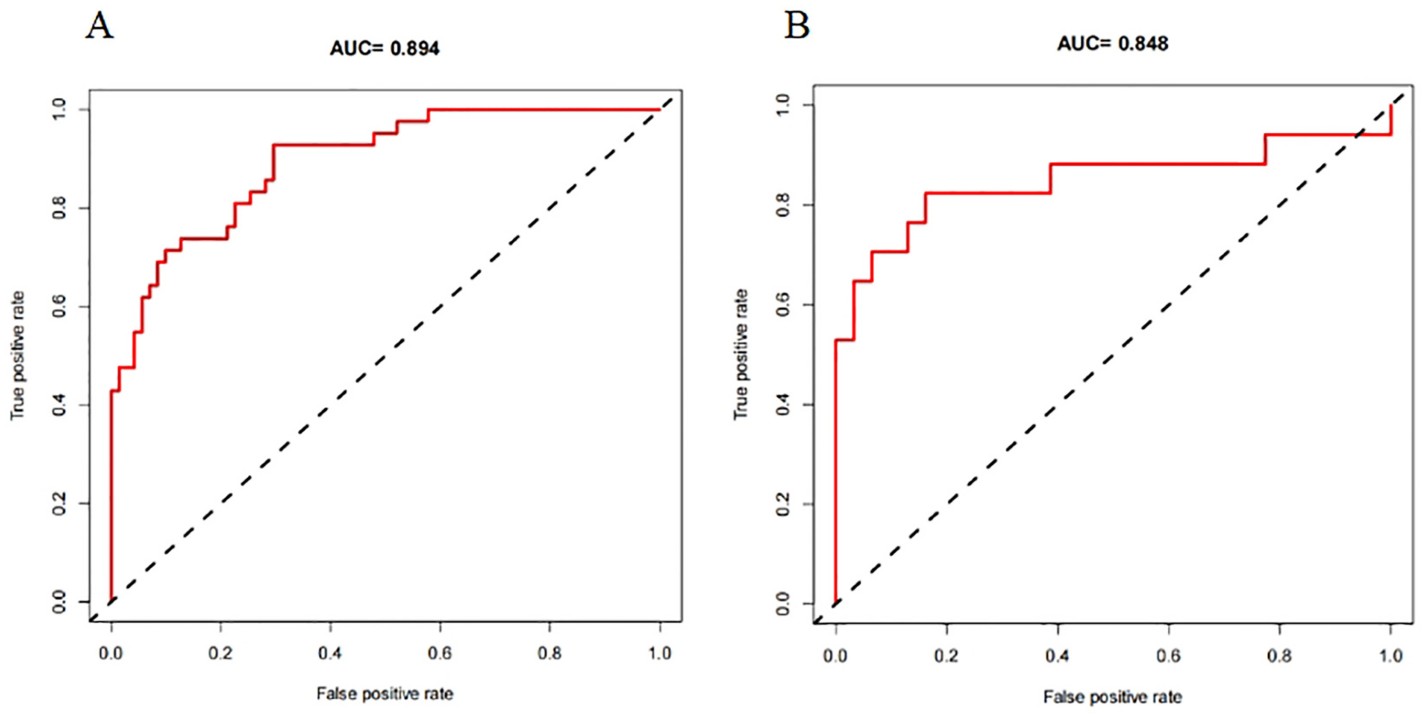

**Figure 5 The receiver operating characteristics (ROCs) curve of the nomogram model to predict poor outcomes after mechanical thrombectomy in the training cohort (A) and validation cohort (B).**

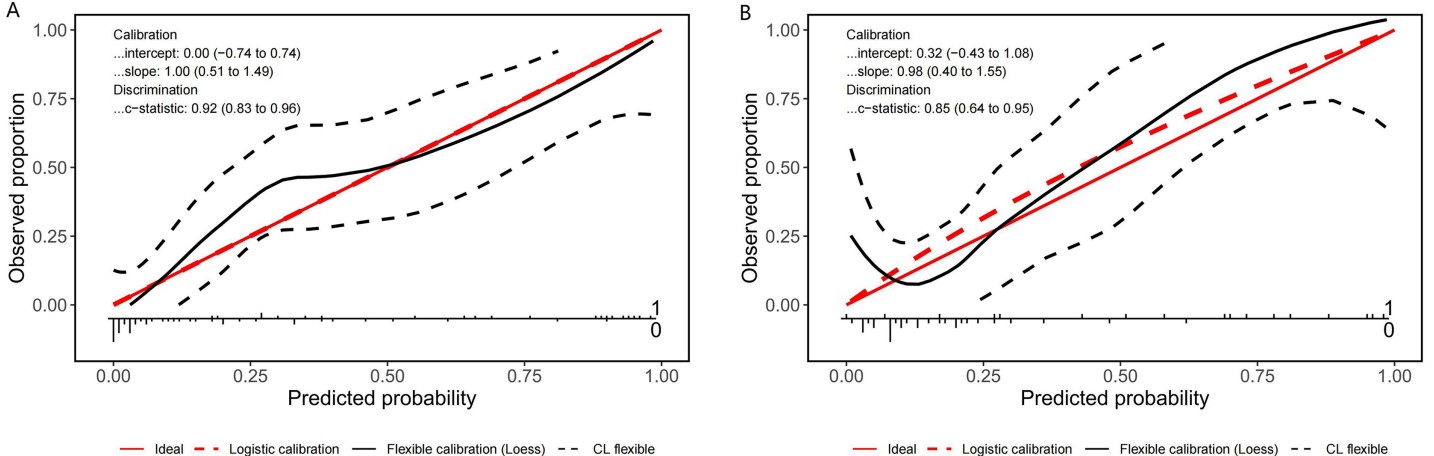

**Figure 6 Calibration plot for predicting poor outcomes after mechanical thrombectomy in the training cohort (A) and the validation cohort (B).**

validation group (Fig. 7B). These findings highlight the strong clinical validity of the nomogram across both the training and validation cohorts. In our study, we utilized imaging data and clinical data to develop imaging models and clinical models, respectively. ASPECTS > 6 and HMCAS were used to construct the imaging model of outcomes of mechanical thrombectomy. The NIHSS score, Wbc, and age were used to construct the clinical model outcomes of mechanical thrombectomy. The findings indicated that the
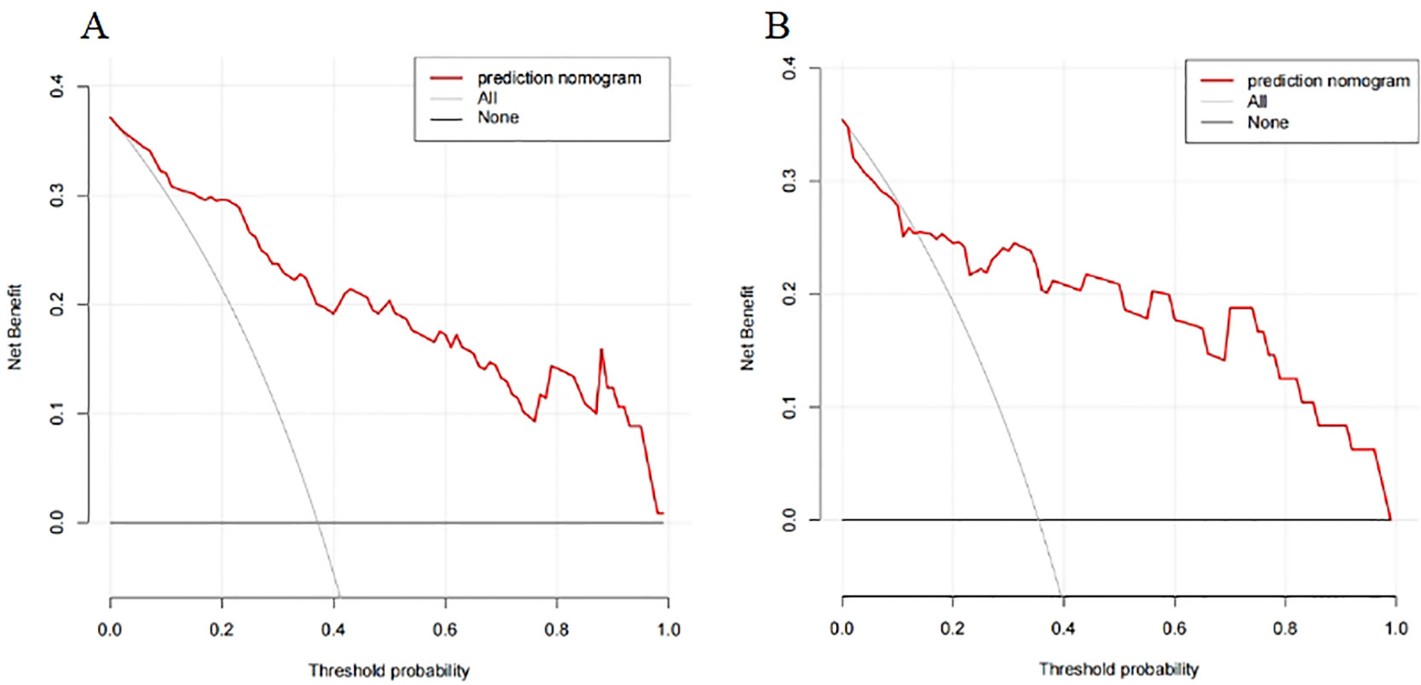

**Figure 7 The DCA of the nomogram predicting poor outcomes after mechanical thrombectomy in the training cohort (A) and the validation cohort (B).**

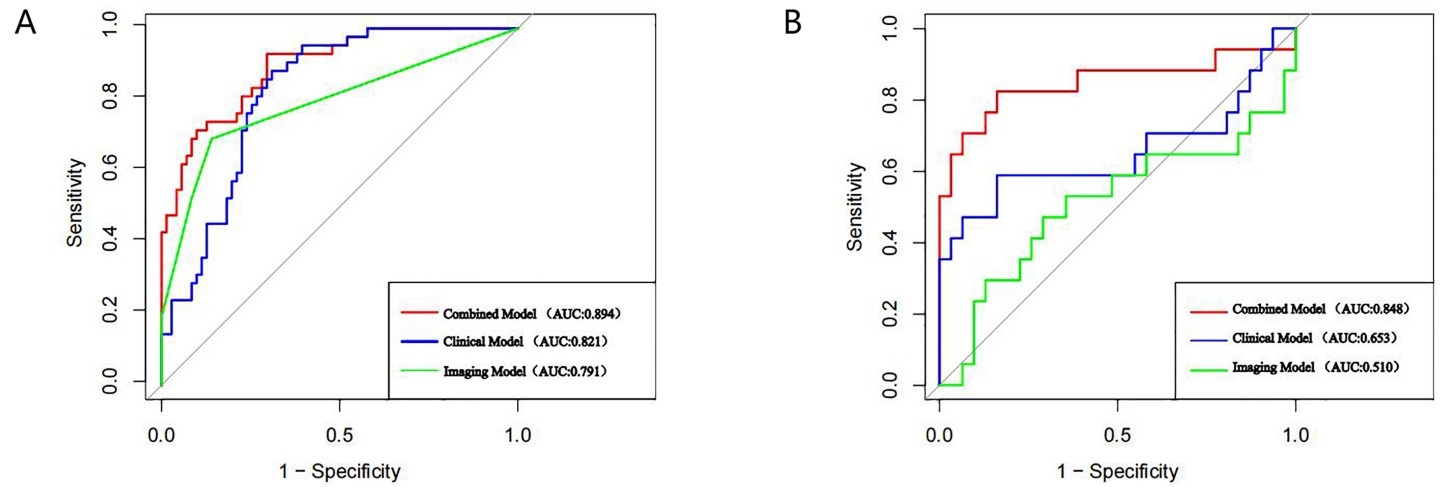

**Figure 8 AUC curves of each model in the training set (A), and validation set (B).**

combine model surpassed both the imaging model and the clinical model in prognostic prediction, exhibiting a superior AUC in both the training (Fig. 8A) and validation sets (Fig. 8B).

We also examined the comparative incidence of stroke in the left and right hemispheres. In our study cohort, the prevalence of left hemisphere stroke was 51.6% ($n = 83$), while the prevalence of right hemisphere stroke was 48.4% ($n = 78$) among all patients. We computed the AUC for patients in the left and right hemispheres independently (Fig. 9).

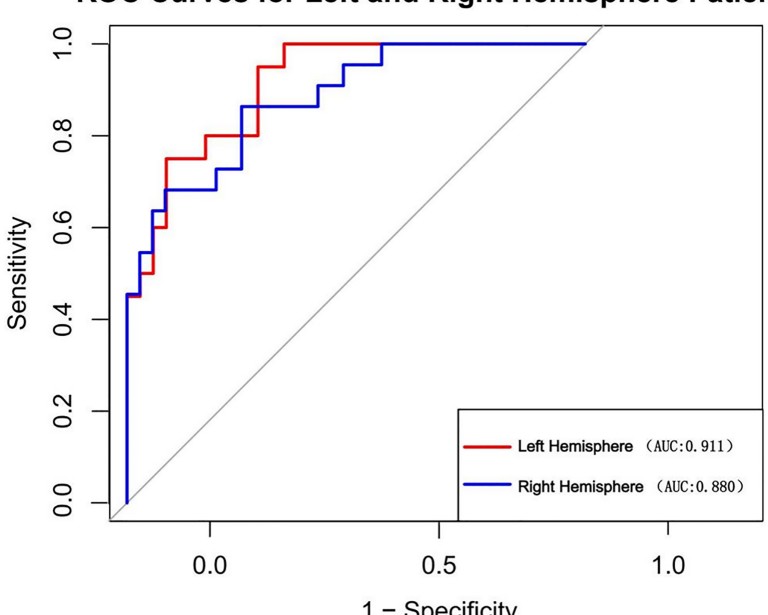

**Figure 9 Nomogram's AUC for Left *vs*. right hemisphere patients.**

The AUC was 0.911 for patients with left hemisphere involvement and 0.880 for those with right hemisphere involvement. The DeLong test *p*-value for the disparity between these AUCs is 0.589, signifying that the discrepancy lacks statistical significance.

## DISCUSSION

We performed retrospective single-center study to develop and validate a practical nomogram based on five predictive factors: WBC, NIHSS score, HMCAS, ASPECTS > 6, age. *Li et al. (2021)* utilized age, NIHSS score, and creatinine as independent risk variables for poor outcomes during mechanical thrombectomy and developed a nomogram model. The model's AUC was 0.816 (95% CI [0.762–0.871]).

In previous studies, mRS at 90 days is the most commonly used primary outcome measure in stroke treatment trials. Due to the necessity of an extended follow-up duration, numerous research are seeking a more rapid and convenient approach to surrogate endpoint indications. A study found the significant correlation between NIHSS score and 90-day mRS score (*Chalos et al., 2020*). The influence of endovascular treatment on mRS was predominantly mediated by the NIHSS. The NIHSS score at 1 week qualifies as a surrogate endpoint and can serve as a primary outcome measure in acute ischemic stroke therapy trials. Another research found that approximately two-thirds of patients exhibited excellent outcomes when their NIHSS score was less than or equal to 3 on day 7; conversely, few patients with baseline scores surpassing 15 attained favorable outcomes after 3 months (*Kasner, 2006*). Therefore, we have set an NIHSS score of 15 as the threshold for binary outcomes. A study indicated that, out of the 42 possible points on the NIHSS score, seven points are directly related to language assessment (orientation

questions, 2; instructions, 2; aphasia, 3), whereas only two points are linked to neglect. The left hemisphere is the predominant language hemisphere in 99% of right-handed individuals (comprising 90% to 95% of the population) and 60% of left-handed individuals. Consequently, it is speculated that the NIHSS may evaluate the severity and extent of strokes in the right hemisphere differently than in the left hemisphere (*Woo et al., 1999*). We have reviewed our sample and analyzed the relative incidence of left and right hemisphere strokes. In the training set, the incidence of left hemisphere strokes 48.7%, whereas the incidence of right hemisphere strokes 51.3%. The probability of adverse outcomes in patients with right hemisphere strokes was 52.4%, and in patients with left hemisphere strokes was 47.6%. In the univariate analysis, no statistical significance was found. Compared with previous studies (*Li et al., 2021*), our research added CT features (including ASPECTS and HMCAS) on a clinical basis, hence enhancing predictive capability. The nomogram demonstrated strong predictive performance in both the training cohort (AUC, 0.894; 95% CI [0.8362–0.9518]) and validation cohort (AUC, 0.848; 95% CI [0.7034–0.993]), assisting neurologists in preoperatively assessing the probability of poor outcome in patients after mechanical thrombectomy. Additionally, as confirmed in both cohorts, our nomogram demonstrated outstanding calibration by precisely predicting the probability of poor outcome beforehand surgery. In clinical situations, the DCA showed how well our nomogram predicted unfavorable outcomes following MT.

The strength of our work is that we thoroughly evaluated brain NCCT signs in patients who had experienced an acute stroke, in contrast to earlier studies that usually only included one or a few brain NCCT signs in AIS patients (*Zhang et al., 2023*; *De Brouwer et al., 2020*). Additionally, these five predictors we ultimately acquired are all common examination items that are gained at admission and are quite simple to obtain. So the nomogram is quite useful and broadly applicable. In patients predicted by our model to be at high risk of poor outcomes, physicians may need to consider the need for MT in the context of the individual patient's situation.

Similar to the majority of earlier research, we also discovered that age, NIHSS score, and ASPECTS are significant influencing factors for a poor prognosis in patients with acute stroke (*Mokin et al., 2018*; *Olivot et al., 2022*). Older people need longer recovery times following surgery because they often have more underlying medical conditions than younger patients do, including hypertension, atrial fibrillation, atherosclerosis, and other conditions. Additionally, elderly patients are more likely to suffer complications, which could affect their ability to recover. This was further supported by a recent large-scale real-world investigation (*Beuker et al., 2023*), which discovered that patients 80 years of age and older were more likely than those under 80 to experience moderate to severe handicap. At 1 year, less than 25% of patients experienced a favorable outcome. The NIHSS score and the ASPECTS are standardized tools used to assess the severity of the disease in AIS patients. The patient's neurological deficit is more severe and the prognosis is often worse the higher the NIHSS score (*Saber & Saver, 2020*). The ASPECT scoring system is widely used to evaluate the degree of early infarction in stroke patients. It has been demonstrated to be a valid predictor of clinical outcomes. Patients with ASPECTS of 0–5 were examined by *Mourand et al. (2018)*, who discovered that those under 70 had a higher chance of

benefiting from endovascular therapy. Patients with ASPECTS ratings of 0–5 and 6–10 were compared in another study (*Kaesmacher et al., 2019*), and the results showed that although these patients can still benefit from reperfusion, individuals with 0–5 scores had a reduced surgical recanalization rate and a worse prognosis. It is unreasonablel to restrict endovascular therapy to younger patients with ASPECTS 0–5, given that the age difference was not statistically significant. The question of whether MT should be performed on patients with poor ASPECTS is still up for debate. It might still require a thorough consideration depending on a number of variables.

HMCAS was defined if the lumen of middle cerebral artery appeared more dense than adjacent or equivalent contralateral arteries but non-calcified on pretreatment NCCT. Research has verified that its manifestation is associated with thrombotic vascular occlusion (*Kang et al., 2023*). This imaging symptom is typical and specific for early ischemic cerebral infarction. Numerous investigations have verified that HMCAS is linked to unfavorable consequences following thrombolysis (*Kim et al., 2017*); nevertheless, its influence on mechanical thrombectomy is still unknown. Recent studies (*Hong et al., 2022*) have found that the sign of HMCAS is directly related to poor leptomeningeal collateral circulation and will lead to worse clinical outcomes. This may be the case since larger thrombi and more serious neurological issues are frequently associated with HMCAS. Larger thrombi have the potential to seriously damage tissues by obstructing blood flow in bypass arteries. *Viltuznik et al. (2021)* found that the average density value of the occluded middle cerebral artery is related to clinical outcomes. The density of the thrombus is thought to be the primary factor influencing mRS evaluation, and the thicker the thrombus, the higher the average density (low serum content and older clots). On the other hand, individuals with HMCAS will require more thrombectomy procedures and more time overall (*Chen et al., 2023*). These may be the reasons why AIS patients with signs of HMCAS have worse clinical outcomes.

Neurological dysfunction and ischemic brain injury can be made worse by inflammatory responses. As vital immune cells in the human body, white blood cells are typically employed to assess the body's level of inflammation and immunological state. According to a number of studies, the severity, prognosis, and mortality of a stroke may be related to an elevated white blood cell count at admission (*Zheng et al., 2018*; *Vo et al., 2023*). Elevated white blood cell counts in individuals experiencing acute stroke may indicate an intensification of the body's inflammatory reaction, leading to the worsening of cerebral ischemia-reperfusion injury and neuronal cell death. Furthermore, the excessive rise in white blood cells can also trigger the secretion of chemicals that promote blood clotting and inflammation, so worsening the blockage of blood vessels in the brain and causing more severe damage from lack of blood flow, ultimately worsening the stroke. Clinically, 'no reflow' often occurs when the blocked blood vessel is completely reopened, but the restoration of blood flow to the brain tissue is incomplete. Studies (*Langhauser et al., 2012*) have demonstrated that the interplay between white blood cells, platelets, and endothelial cells can lead to tissue harm after temporary interruption of blood flow to the brain and subsequent restoration of blood supply. *El Amki et al. (2020)* showed that

stagnant cerebral blood flow is caused by neutrophils that are trapped in capillaries, and that treatment targeting neutrophils can improve the flow of blood in small blood vessels. *Malhotra et al. (2018)* discovered that increased levels of neutrophils and white blood cells were linked to poorer clinical outcomes in patients with acute ischemic stroke (AIS). Thus, it is imperative to enhance the anti-inflammatory therapy during the initial phases for individuals with AIS.

The current research still has certain limitations. Firstly, it is a single-center retrospective analysis with a small sample size. While the model shows good discriminant performance, it lacks external validation. Before clinical application, a multi-center study is necessary to validate its performance. Secondly, although our study included additional variables, there may still be some risk factors associated with poor outcomes after MT that were not considered. Thirdly, due to the selection of our target population, further investigation is required to determine whether this model can be widely applied to the stroke population. Fourth, our model encompasses an age range of 30 to 95 years, with an NIHSS score upon admission ranging from 0 to 35. Exercise caution while utilizing this nomogram for patients beyond this range. Furthermore, like many retrospective studies, our research is susceptible to selection bias, as we excluded patients with missing clinical data. To address this limitation, we have provided detailed baseline information about the patients to aid other institutions or researchers in using or comparing our models.

## CONCLUSIONS

Our study introduces a novel and practical nomogram that utilizes HMCAS, ASPECTS > 6, NIHSS score, age, and WBC to effectively predict the risk of poor outcomes following mechanical thrombectomy in patients with acute ischemic stroke.

### Funding

This study was supported by the Science and Technology Program of Guangzhou, China (202102080665), and the Huadu district academy of basic and applied basic research co-funded the project (23HDQYLH16, 23HDQYLH17). The funders had no role in study design, data collection and analysis, decision to publish, or preparation of the manuscript.

### Grant Disclosures

The following grant information was disclosed by the authors:
Science and Technology Program of Guangzhou, China: 202102080665.
Huadu district academy of basic and applied basic research: 23HDQYLH16, 23HDQYLH17.

### Competing Interests

The authors declare that they have no competing interests.

## Author Contributions

- Jingyao Yang conceived and designed the experiments, performed the experiments, authored or reviewed drafts of the article, and approved the final draft.
- Fangfang Deng performed the experiments, prepared figures and/or tables, and approved the final draft.
- Qian Zhang performed the experiments, prepared figures and/or tables, and approved the final draft.
- Zhuyin Zhang analyzed the data, prepared figures and/or tables, and approved the final draft.
- Qinghua Luo analyzed the data, prepared figures and/or tables, and approved the final draft.
- Yeyu Xiao conceived and designed the experiments, authored or reviewed drafts of the article, and approved the final draft.

## Ethics

The following information was supplied relating to ethical approvals (*i.e.*, approving body and any reference numbers):

Ethical Application Ref: Guangzhou Hospital of Integrated Traditional and Western Medicine 20240325003.

## Data Availability

The raw data are available in the Supplemental Files.

## Supplemental Information

Supplemental information for this article can be found online at http://dx.doi.org/10.7717/peerj.18662#supplemental-information.

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
