# Peer review of "A novel nomogram based on the patient’s clinical data and CT signs to predict poor outcomes in AIS patients"

_PeerJ, doi:10.7717/peerj.18662_

## Round 0.1 · original submission · Major Revisions

The article under review will be of interest to the journal's readership. However, there are several issues that need to be addressed before it is accepted. The authors are requested to carefully address the reviewers' comments and resubmit a revised manuscript incorporating the necessary changes.

·

Basic reporting

The manuscript entitled “A novel nomogram based on the patient's clinical data and CT signs to predict poor outcomes in AIS patients” develops a tool for predicting poor post-thrombectomy outcomes in ischemic stroke patients. This topic is relevant, but there are several key issues with the methods and reporting which need to be addressed before this project is ready for publication.

1. More detail about how to use the proposed nomogram is needed. For example, it is unclear how each of the individual measures are combined to yield the total score. The results section reports that each measure corresponds to a score of between 1-100, but wouldn’t this lead to a total score of 500 rather than 240 (as reported in Figure 3). Why are different measures worth different numbers of points and how was this weighting established? Given that NIHSS scores range from 0-42, why is the maximum possible NIHSS score listed as 35? Overall, the description of the proposed nomogram needs to be expanded to include explicit instructions on how this tool is intended to be used, how total scores are generated, and what methods were used to establish the score weights used in the tool.
2. More detail is needed in the description of the LASSO analysis. Was the collinearity of variables assessed before LASSO variable selection or after? In cases where input variables carry similar information, LASSO will arbitrarily select one variable and eliminate another. This means the covariates selected (and their associated weights) may differ when LASSO analysis is repeated. How was this issue addressed? Why was the training/testing data split in a 7:3 ratio? This ratio is not standard so it would be useful for the authors to provide a justification of why it was used. It would also be useful to include some tests to evaluate whether your model is overfit. For example, does the generated model remain similar in terms of covariates included and accuracy in the test set when different patients are included in the training/testing sets?
3. Please provide additional details on how each of the considered variables were quantified. Specifically, starting on line 97 the authors list a series of considered variables. Some of these variables (e.g. NIHSS scores, ASPECTS) correspond to clearly defined measures while others could be quantified using several different measures (e.g. atrophy, calcification, Leukoriasis). What measures were used to quantify these variables?
4. It would be beneficial if much of the discussion content was introduced earlier in the manuscript (i.e. in the introduction). For example, it would be helpful to have a more detailed description of relevant past work (and key knowledge gaps) in the introduction.
5. It is important to discuss potential generalisability issues in the limitations section. This tool was built using data from a non-representative subset of the stroke population (e.g. MCA stroke only, no haemorrhages, only patients referred for thrombectomy). It will therefore be important to explicitly acknowledge that this tool cannot be assumed to be accurate in the full stroke population.
6. Please add a visualisation of confidence interval to figure 5. This is important because there are a very low number of cases which occur in the higher end of the nomogram-predicted probabilities. Calibration plots are generally not accurate when such small numbers are used, which is why probability bins are normally used. Given the small number of cases included in the test set, I would expect that the calibration plot’s estimates would be very uncertain.
7. Similarly, figure 2 would benefit from a clearer description. Please provide a key defining each line colour and axis measurement. It is also important to include a more detailed description of what these plots show, as many readers will not be familiar with LASSO plots.
8. There are many acronyms used in this manuscript, many of which are not defined the first time they are used. For example, HMACS is used throughout the paper, but isn’t defined until the introduction. I think reducing the number of acronyms to a minimum would greatly improve manuscript readability.

Experimental design

1. This study has binarized good/poor outcomes based on post-operation NIHSS scores. While this approach is relatively standard, I think it may be more informative to consider NIHSS score changes relative to baseline. For example, using the current approach a patient with a baseline NIHSS of 30 and a post-op NIHSS of 16 and a patent with baseline 0 and post-op of 16 would both be classed as poor outcomes but represent qualitatively different scenarios. Given the high correlation between pre-op NIHSS and post-op NIHSS, it would be more helpful to have a tool which will distinguish between people who will get worse or better rather than a tool which predicts total score. Does your nomogram perform similarly when outcome is quantified as change relative to baseline NIHSS? Please include additional analyses to investigate this issue.


2. Similarly, why is the ASPECTS score binarized rather than treating it as a continuous variable? I understand that previous studies have binarized ASPECTS in this way, but I would be surprised if binarizing this measure provided more information that treating it as a continuous variable. Does considering ASPECTS as a continuous variable change your results in terms of model fit or prediction accuracy?


3. I think it will be important to investigate potential differences between patients with left and right hemisphere lesions. The NIHSS tends to assign higher scores to left hemisphere patients relative to right hemisphere patients in cases where the underlying strokes are identical in size (e.g. Woo et al., 1999). This could mean that left hemisphere patients are disproportionally likely to be classed as poor outcomes in your study. Please report the relative incidence of left and right hemisphere strokes in your sample. Is your nomogram’s AUC similar in left vs. right hemisphere patients? Does considering stroke hemisphere improve the fit of your models?
4. Please justify your inclusion/exclusion criteria. Why was hospital stay < 7 days an exclusion criterion? Why were patients with haemorrhages excluded? Why was the sample restricted to just an MCA cohort?

Validity of the findings

1. The authors have demonstrated that their proposed tool is able to help distinguish between patients with good/poor NIHSS scores. However, the extent to which the proposed composite score outperforms the prognostic information provided by the individual predictors in isolation. For example, is the nomogram AUC significantly better than the AUC when outcomes are predicted from baseline NIHSS or baseline ASPECTS alone? This is an important question to explore as it is important to illustrate that new prognostic measures outperform existing and more easily accessible prognostic measures.

Additional comments

1. “The benefits of NCCT to MRI include ease of use, affordability, speedy imaging, and an easy-to-see representation of anatomical components” – I agree that CT is very useful, but it’s ability to visualise anatomy is very poor compared to many standard MRI sequences. Please revise this claim accordingly.
2. “Although MT therapy is beneficial for patients with large vessel occlusion strokes, nearly half of these patients had worse prognoses and a greater 90-day mortality rate, according to published research” – Please provide a bit more detail here. What comparison is used to determine “worse prognoses”? Could patients who have thrombectomy just have worse outcomes overall because patients with mild strokes (e.g. lacunar) are less likely to be treated using this method?
3. “evaluated brain NCCT signals”- The term “signs” would be more appropriate that “signals” here as “brain signals” seems to imply functional measures.
4. “addition to modifying treatment plans and enhancing acute stroke patients' prognosis, it can somewhat lessen the cost on society - please explain this claim. How, specifically, does your tool improve prognosis? How does it reduce the cost to society?
5. Do you think the tool is generalisable to patients who don’t undergo mechanical thrombectomy?

·

Basic reporting

The authors have used a well-defined dataset to study a nomogram for predicting poor outcomes in AIS patients. These are my comments.
Basic reporting:
1. The choice of the measure and threshold between good vs poor outcomes needs more explanation. The authors have cited some literature, but not motivated these discussions which are essential to their work. In particular, how does the distribution of the measure used here (i.e, NIHSS score) split between the two classes? A visualization in this regard is necessary.

2. Preprocessing: Though VIF preprocessing is commendable, the use of tolerances is not clear and needs definition and expression(s).

3. English translation of the Ethics application and approval documents maybe necessary in line with PeerJ policy

4. HMCAS abbreviation needs to be explained where it is first used.

Experimental design

Code is necessary to replicate and use the findings, in line with PeerJ policy for Bioinformatics manuscripts. A webserver may also be useful.

Validity of the findings

1. Validation: estimation of the bootstrapped c-index of the nomogram is recommended.
2. The absence of significance of chi-squared tests needs justification in the context of the study. "cohort (X2= 5.010, df =8, P =0.756) and the validation cohort (X2= 5.799, df =7, P =0.563)"
3. The signature of the nomogram consists of two kinds of features:
i. those with high odds-ratio and high significance
ii. those with low odds and low significance
--> The authors may comment on such a fallout from the analysis, and if the model is robust to elimination of the second class of features indicated above.

Additional comments

On the whole, the study is promising, but needs a bit more work for recommendation.

---

## Round 0.2 · Minor Revisions

The authors are requested to address the additional set of comments provided by the reviewer and especially include most of the comments provided in the rebuttal in the main manuscript.

·

Basic reporting

1) In several cases, the authors have responded to reviewer questions in the response document, but have not included the suggested changes in the manuscript. I have highlighted the most important instances of this below. Please address these comments/questions in the manuscript.

o Basic Reporting comment 1: The authors have answered my query in the response, but this information needs to be made more explicit in the text as well. Please add more detailed instructions on how to use the nomogram to the manuscript.

o Basic Reporting comment 3: Please include a more detailed description of how each variable was quantified in the manuscript text. This is critically important because the nomogram can only be used in cases where each of these variables has been quantified in the same way that you did in your study.
o Experimental design comment 1: Given that both reviewers enquired about why NIHSS scores for outcomes, it is important to add an explanation of why this approach was taken in the main text, not just in the reply letter.
o Experimental design comment 3: Please add this requested analysis to the manuscript draft.
o Additional comments 2, 4, 5: please include a discussion of these potential issues/topics in the manuscript discussion.

Experimental design

1) Regarding Basic Reporting Comment 2: VIF/Colinearity should be assessed prior to variable inclusion in LASSO analysis. LASSO is not suitable for cases where input variables exhibit severe collinearity and this needs to be checked before the analysis is conducted, not after. This is a critical issue which must be addressed.

2) With regards to my previous enquiry about why NIHSS scores >35 weren’t included in the nomogram (Basic Reporting Comment 1), I now understand that scores >35 were not included in the nomogram as these high scores were not present in the training sample. If this is the case, this approach needs to be consistent across all variables included in the nomogram. For example, the nomogram includes the age range 30-95, but the study sample only includes patients between the age of 34 and 92. Please revise this measure to only include ages which were included in the study sample. It should also be explicitly be mentioned in the limitations that the nomogram cannot be used in patients outside this age range or with NIHSS scores >35.

Validity of the findings

See above comments.

·

Basic reporting

-nil-

Experimental design

-nil-

Validity of the findings

-nil-

Additional comments

The authors have addressed all my previous comments, and I have no further suggestions, heartily recommend the same.

---

## Round 0.3 · accepted · Accept

All the reviewers have acknowledged that their concerns have been addressed. Congratulations on the article's acceptance.

·

Basic reporting

The authors have adequately addressed all of my comments.

Experimental design

N/A

Validity of the findings

N/A

Additional comments

N/A